# Transcultural Differences in Risk Factors and in Triggering Reasons of Suicidal and Self-Harming Behaviour in Young People with and without a Migration Background

**DOI:** 10.3390/ijerph17186498

**Published:** 2020-09-07

**Authors:** Zeliha Özlü-Erkilic, Thomas Wenzel, Oswald D. Kothgassner, Türkan Akkaya-Kalayci

**Affiliations:** 1Outpatient Clinic of Transcultural Psychiatry and Migration Induced Disorders in Childhood and Adolescence, Department of Child and Adolescent Psychiatry, Medical University of Vienna, 1090 Vienna, Austria; zeliha@gmx.at; 2Postgraduate University Program Transcultural Medicine and Diversity Care, Medical University of Vienna, 1090 Vienna, Austria; 3Department of Psychiatry and Psychotherapy, Medical University of Vienna, 1090 Vienna, Austria; thomas.wenzel@meduniwien.ac.at; 4Department of Child and Adolescent Psychiatry, Medical University of Vienna, 1090 Vienna, Austria; oswald.kothgassner@meduniwien.ac.at

**Keywords:** suicide attempt, self-harming, emergency psychiatry, transcultural differences, migration background, risk factors, triggering reasons

## Abstract

Minors with and without migration background can have different risk factors and triggering reasons for self-harming and suicidal behaviour. We retrospectively analysed the data of 192 children and adolescents to investigate the transcultural differences in self-harming, as well as suicidal behaviour in Austrian, Turkish, and Bosnian/Croatian/Serbian (BCS)-speaking patients, who were treated in an emergency out-patient clinic in Vienna. Our results showed transcultural differences in both behaviours. In all groups, females had higher rates of suicide attempts and self-harming behaviour than males. While Turkish-speaking patients received treatment more often, after attempted suicide, Austrians and BCS-speaking patients needed treatment more often for acute stress disorder. Suicide attempts and self-harming behaviours were triggered most frequently by intrafamilial problems, but more frequently in migrant patients. Turkish-speaking patients were at a more than 2 times (OR = 2.21, 95%CI: 1.408–3.477) higher risk for suicide attempts, and were triggered almost 3 times (OR = 2.94, 95%CI: 1.632–5.304) more often by interfamilial conflicts. The suicide attempts of BCS-speaking minors were more often caused by relationship and separation crises (OR = 2.56, 95%CI: 1.148–5.705). These transcultural differences in suicidal and self-harming behaviour of minors, demand an increase of transcultural competence to provide optimal treatment of migrant children.

## 1. Introduction

According to the World Health Organisation, worldwide about 800,000 people die due to suicide each year [1]. While the risk of dying from completed suicide rises with increasing age, suicide attempts are statistically more common among adolescents [2]. Additionally, auto-aggressive behaviours such as self-harming [3] and suicidal behaviour [2] are associated with younger age. Numerous studies reported that suicidality is the most frequent reason for a referral to paediatric Emergency Departments [4,5,6].

### 1.1. Self-harming and Suicide Attempt

Self-harming behaviour or “non-suicidal self-injury” is defined as direct and deliberate destruction of body tissue, with no intent to die, e.g., skin cutting [7]. Whereas a suicide attempt is a self-injurious behaviour in which the aim is to die, e.g., shooting oneself [8]. Suicidal ideation is defined as “thoughts of engaging in behaviour intended to end one’s life” [9].

### 1.2. Distinguishing Suicidal Ideation from Suicide Attempt

The “Integrated Motivational-Volitional Model” (IMV) maintains that suicide is a behaviour resulting from different factors and describes the crucial aspects from ideation to behaviour in detail.

The IMV model describes three stages of suicidal behaviour—(1) pre-motivational phase (background factors that might cause suicide); (2) motivational phase (development of suicidal ideation) and (3) volitional phase (factors crucial for transition from ideation to attempting suicide). The volitional phase factors, e.g., exposure to suicide, access to means, planning, impulsivity, physical pain sensitivity, fearlessness about death, imagery and past behaviour, are crucial for transforming suicide ideations to suicide attempts. The study of Branley-Bell et al. [10] compared suicide attempters with those showing suicide ideation, in the three stages of suicidal behaviour, as mentioned above. In the motivational phase, there were no differences between suicide attempters and those showing suicidal ideation, but the volitional phase factors of these two groups varied. In all volitional phase factors, suicide attempters varied from the ideation group—they reported a higher capability for suicide, were more impulsive, and more frequently had a family member or friend who had self-injured or attempted suicide [10].

### 1.3. Risk Factors

Although suicide attempts are more frequent among females [11,12,13,14,15], more males [16,17] die by suicide. Young women are more exposed to risk factors that can lead to suicide attempts, and their vulnerability is also higher, compared to male adolescents [18]. In addition, diverse genetic, biological, psychiatric, psychological, social and cultural factors can influence suicidal as well as self-harming behaviour [19]. There is a considerably larger range of specific factors that increase the risk of suicidal behaviour of youths, such as—previous suicide attempts [20] and acts of self-harm [21], adverse family experiences [22], parental hostility [23], parental violence [19], psychological abuse [24], low peer-group integration [25], emotionally stressful experiences [26], adverse personality traits [27], negative life events [28], low and ineffective coping strategies [29], mental disorders [30], financial problems in the family [25], substance abuse [31], parental addiction [32], interpersonal problems [33], school-related stress [34] and hopelessness [34].

Similarly, self-harming behaviour might be associated with risk factors like sexual and physical abuse in childhood [35], low self-esteem [35], social isolation [36], impulsivity [37], hopelessness [38], bullying [37], mental disorders [30], as well as perfectionism and self-criticism [39]. Certainly, negative experiences within the family, like maladaptive parenting [40], child and family adversity including parental divorce [35], poor parent–child attachment [41], and low socioeconomic situation of the family [42], can also evoke self-harming behaviour.

### 1.4. Transcultural Risk Factors

Having a migration background or migration experience could also be expected to be risk factors for different forms of suicidal behaviour [43]. Suicidal behaviours in different migrant groups are higher [44] not only because of cross-cultural differences in their risk and resilience factors [45] but because they also have a higher lifetime rate of self-harming behaviour and attempted suicides [46], compared to their native peers.

Migration-induced stressors such as language difficulties, loss of social networks, identity problems, discrimination, racism, discrepancies between norms and values of the society of origin, and that of the host society, living conditions in the host country, problems of acculturation, and stressful life events can negatively affect youths with a migration background [47]. In particular, self-harming and suicidal behaviour arise more often in adolescents with low socioeconomic status [48] and with more experiences of negative life-events [49]. Migrants living in Austria, like those in other European countries, have lower income levels and consequently have a much higher risk of poverty, compared to the indigenous population [50]. Therefore, migrant populations often have a higher rate of psychological problems [51].

The objective of the present study was to explore transcultural differences in risk factors and in triggering reasons of suicidal and self-harming behaviour, by comparing young outpatients with and without a migration background in a clinical population. We further wanted to evaluate the observation reported in most international studies, including those mentioned above, that suicide attempts and self-harming behaviour are the most frequent reasons for an acute referral, when analysing referrals in both subgroups in Austria. We also hypothesised that different transcultural risk factors could be observed to contribute to suicidal and self-harming behaviour among children and adolescents, with and without a migration background, as relevant key factors to be considered in prevention and as a possible treatment focus. To our best knowledge, no study in Europe up until now explicitly analysed the transcultural differences in self-harming and suicidal behaviours among children and adolescents, with and without migration backgrounds, such as those most common in Austria (Austrian and Turkish as well as Bosnian/Croatian/Serbian-speaking (BCS-speaking).

## 2. Material and Methods

### 2.1. Participants

Over a three-year period, a total of 1821 children and adolescents were admitted to the Emergency Unit of the Department of Child and Adolescent Psychiatry at the Medical University of Vienna. For the present study, which was conducted between 06/2011–06/2014, in the first-step evaluation process we included the records of 1718 patients aged between 4–18 years old, as 103 of the patients did not fit into the age range of our study. In the second step, only the data of 1093 patients belonging to three nationalities—Austrians (*n* = 800), BCS-speaking (*n* = 163), and Turkish-speakers (*n* = 130) were analysed for their reason of admission to the emergency psychiatric clinic. Afterwards, the total data of 192 children and adolescents of these three groups who showed self-harming behaviour (*n* = 47) and attempted suicide (*n* = 145), were analysed in detail, for the risk factors and triggering reasons. Patients of other nationalities were excluded, as the present study focused on the transcultural differences between Austrian, Turkish, and BCS-speaking patients.

After Germans, the BCS and Turkish-speaking migrants are the largest migrant communities living in Austria [52], therefore, in the present study, we concentrated on these two migrant groups.

In the present study, the more specific ethnic origin of the patients from Turkey (such as “Kurdish” and “Armenian” of the wide range of ethnic groups living in Turkey) was not part of the records available. Therefore, we use the term “Turkish-speaking minors” for all minors originating from Turkey, which does not refer to their mother language, or language predominantly used in everyday life, or to primary citizenship. Therefore, some participants might still speak a different mother tongue in daily life than Turkish, and have different ethnic backgrounds as well (e.g., Arab, Kurdish, etc., or in second-generation patients, German) and different present official or dual citizenships (Turkish, Austrian, or other).

The definition of “migration background” used in this study follows the Austrian public statistical standards definition, which states, “at least one parent or grandparent was not born in the host country” [50].

Two clinics offer medical care for the mental-health problems of children and adolescents living in Vienna. One of these two clinics, the Department of Child and Adolescent Psychiatry of the Medical University of Vienna, offers medical care for about half of the Viennese children and adolescents aged between 0–18 years. Each clinic is responsible and offers medical care only for the assigned districts. The migrant populations vary considerably in these different districts, in some districts, the number of migrants is quite high, but in others, it is quite low. Therefore, we suppose that these differences provide a realistic sample composition for patients with and without a migration background. As the Department of Child and Adolescent Psychiatry of the Medical University of Vienna is responsible for the psychiatric treatment of more than half of the Viennese children and adolescents with mental health problems, our study sample is therefore representative of the Viennese population.

### 2.2. Procedures and Measures

We applied a questionnaire (standardised data matrix) with a specific data sheet developed for this study, to be used for the retrospective data analysis, which included items from the hospital standard medical history, socio-demographic data like nationality, country of origin, languages spoken at home, age, and sex, reasons for referral, psychosocial details and migration-specific information. Furthermore, we analysed different risk factors like migration background, problems with peers, problems at school, etc., as identified in the earlier research outlined before, which might particularly influence the suicidal and self-harming behaviour of the youths. In particular, we focused on the transcultural differences in suicidal and self-harming behaviour, between the three study groups, as well as separately within each study group.

Our results are therefore based on the standardised file records. During the acute or emergency treatment of children and adolescents, the anamnesis (medical history) might not necessarily be recorded in full detail in the hospital’s records. Consequently, our results are based on sometimes incomplete sets of data for each case.

We used IBM SPSS Statistics, version 24 to perform the statistics evaluation [53]. Odds ratios (OR) were calculated to capture the impact of the risk factors. We compared each specific group in relation to other groups using ORs. The OR is the ratio of the odds obtained from two different dichotomous variables, using a formula reported by Bland et al. [54]. For significance testing, an alpha level of 0.05 was applied. Some results with an alpha level of 0.10 are displayed with the tendency to significance.

The present study protocol was approved by the Ethics Committee of the Medical University of Vienna (Registration number: EK 901/2010).

## 3. Results

In the first-step of the evaluation process, the records of 1718 patients aged between 4–18 years old were analysed. More than half (53.4%, *n*= 918) of them had a migration background and just 46.6% (*n*= 800) of them were native Austrians. A total of 9.5% (*n* = 163) of the patients with migration background originated from Bosnia/Croatia/Serbia, 7.6% (*n* = 130) were from Turkey, and 36.4% (*n* = 625) were originally from other countries like India, Pakistan, Iran, Iraq, Afghanistan, Egypt, etc. Based on the already listed criteria, the present study included 1093 files of patients from Austria (*n* = 800), from Bosnia/Croatia/Serbia (*n* = 163), and from Turkey (*n* = 130), for statistical analyses of gender and referral reasons to emergency psychiatric outpatient-clinic.

### 3.1. Study Sample

As Table 1 indicates, more female patients than male patients received acute treatment at the emergency outpatient clinic in the Department of Child and Adolescent Psychiatry. This was true for all three investigated groups. The highest gender difference was observed among Turkish-speaking patients, nearly three quarters (71.5%) of them being females.

The mean age of the whole study sample was 14.56 (SD = 3.0, range 4–18) years. The mean age of both genders (females—14.97 years and males—13.97 years), as well as the mean age of three study groups (Turkish-speaking patients—15.11 years, Austrian patients—14.56 years, and BCS-speaking patients—14.14 years) were quite similar. The majority (83.6%) of the patients were aged between 12–18 years, i.e., in puberty.

### 3.2. Referral Reasons in the Different Patient Groups at Admission Time

In the whole study sample, suicide attempts (13.3%) were the third most frequent reason for documented treatment, after acute stress disorder (20.3%) and behavioural disorders (13.9%). Turkish-speaking patients were most likely to receive acute treatment at the emergency outpatient clinic for attempted suicide (23.1%), whereas Austrian (20.9%) and BCS-speaking (19%) patients most frequently received treatment for acute stress disorder.

The number of patients seeking treatment for self-harming behaviour was quite similar between the patient groups. Slightly more Turkish-speaking patients (4.6%) received acute treatment due to self-harming behaviour, as compared to native (4.4%) and BCS background (3.7%) children and adolescents (Table 2).

### 3.3. Gender Differences in Referral Reasons among the Different Patient Groups

Austrian females were most frequently treated at the emergency outpatient clinic due to acute stress disorders (24.4%), while Austrian males were commonly treated for behavioural disorders (26.6%).

BCS-speaking females were admitted most frequently because of a suicide attempt (22.3%), while their countrymen were commonly treated because of a diagnosis of behavioural disorder (23.3%). However, females (23.7%), as well as males (21.6%) originating from Turkey, were most often treated for an attempted suicide at the Emergency department.

Across all three study groups, females had higher rates of suicide attempts and self-harming behaviour than males:

Suicide attempts—females vs. males:Austrian patients: 14.6% vs. 6.3%BCS-speaking patients: 22.3% vs. 6.7%Turkish-speaking patients: 23.7% vs. 21.6%

Self-harming behaviour—females vs. males:Austrian patients: 6.4% vs. 1.7%BCS-speaking patients: 3.9% vs. 3.3%Turkish-speaking patients: 5.4% vs. 2.7%

While males with a migration background had lower rates of suicidal ideation than their fellow countrywomen, among Austrian patients there was almost no gender difference with regards to suicidal ideation:

Suicidal ideation—females vs. males:Austrian patients: 11.8% vs. 12%BCS-speaking patients: 4.9% vs. 3.3%Turkish-speaking patients: 7.5% vs. 0%

### 3.4. Transcultural Differences in Triggering Factors for Suicide Attempts and Self-Harming Behaviour

The data of 145 patients who were admitted to the emergency psychiatric clinic after a suicide attempt and the data of 47 patients who showed self-harming behaviour were analysed in detail for triggering factors. For the present study, we clustered the triggering factors for suicide attempts and self-harming behaviour, according to the information in patient files, as follows—intrafamilial conflicts, relationship/separation crises, trauma, problems at school, psychological disorders, problems with peers, burdens due to physical disorders, and unknown reasons. Table 3 shows the triggering factors of suicide attempt and self-harming behaviour in three patient groups.

In the whole study sample (45.5%) as well as in three study groups, the most frequent triggering factor for a suicide attempt was intrafamilial problems. However, in more than half of the patients with migration background (Turkish—56.7% and BCS—55.6%), suicide attempts were triggered by intrafamilial problems, whereas only 38.6% of the Austrian-patient suicide-attempts were triggered by intrafamilial problems.

Similarly, intrafamilial problems were the most frequently recorded triggering factor for self-harming behaviour. Although no significant differences could be identified concerning triggering factors for self-harming behaviour and gender—in the whole study sample or within the study groups—patients with migration background showed significant gender differences concerning triggering factors for suicide attempts, but this was not the case in Austrians. In all study groups, intrafamilial conflicts were the most frequent triggering reason for attempted suicide, independent of gender, with the exception of Turkish-speaking males. In Turkish-speaking male patients, suicide attempts were most frequently due to relationship/separation crisis (37.5%).

### 3.5. Risk Factors and Triggering Reasons of Suicide Attempt and Self-Harming Behaviour

On the one hand, Turkish-speaking children and adolescents were at a more than 2 times (OR = 2.21, 95 % CI: 1.408–3.477, *p* = 0.001) higher risk for suicide attempts, as compared to other patient groups. Intrafamilial conflicts triggered suicide attempts approximately 2 times (OR = 1.76, 95 % CI: 0.960–3.211, *p* = 0.068) more often in BCS-speaking, and 3 times (OR = 2.94, 95 % CI: 1.632–5.304, *p* = 0.001) more often in Turkish-speaking children/adolescents, than in other study groups. On the other hand, relationship/separation crises triggered suicide attempts 2.5 times (OR = 2.56, 95 % CI: 1.148–5.705, *p* = 0.022) more often among BCS-speaking children/adolescents, and 2 times (OR = 2.12, 95 % CI: 0.846–5.311, *p* = 0.062) more often in Turkish-speaking patients, compared to the other groups in our study. In Turkish-speaking children and adolescents, the risk for a suicide attempt increased about 6-fold (OR = 5.65, 95 % CI: 0.933–34.241, *p* = 0.059) when they had problems with their peers.

Furthermore, compared to other patient groups, the risk for self-harming behaviour among Turkish-speaking children and adolescents was 3 times (OR = 3.19, 95% CI: 0.813–12.484, *p* = 0.096) higher when they had problems at school.

### 3.6. Missing Data

The missing data were different between minors with a migration background and their indigenous peers. For example, among Turkish speaking patients, 6.7% of the triggering factors for the suicide attempts and 5.7% of the Austrian patients were missing. Therefore, we only included records when sufficient main data like nationality, country of origin, age, sex, and reasons for referral and triggering reason for a suicide attempt and self-harming behaviour were mostly available. In some cases, migration-specific information and sociodemographic data were missing, consequently the records of patients with a migration background had more missing data, overall.

## 4. Discussion

The aim of the present study was to identify clinically relevant social and demographic risk factors and triggering reasons for self-harming and suicidal behaviour among children and adolescents, which could be helpful in the diagnosis and treatment of young patients with migration background and in developing preventive measures.

### 4.1. Transcultural Differences among Natives and Migrants

In the hospital records, the percentage of missing data among children and adolescents with a migration background was higher, compared to their native peers. We assumed that especially in an emergency situation, the insufficient language skills of migrants make comprehending their medical history more difficult.

In the present study, the children and adolescents with a migration background were overrepresented in the emergency outpatient clinic. More than half of the children and adolescents treated in the emergency psychiatric outpatient clinic had a migration background, although in 2016, just a quarter (24.3%) of the young population aged between 0–24 years, living in Austria, had a migration background [50]. In line with our results, previous studies showed that adults, as well as minors with a migration background were overrepresented in the emergency outpatient clinics [4,55,56,57]. We assumed that institutional and cultural barriers hinder migrants from using both preventive and regular offers, therefore in Europe, the access of migrant groups to healthcare is noticeably lower than in non-migrant groups [58]. Among children and adolescents with a migration background, the prevalence of emergency psychiatric problems due to insufficient early interventions is higher, compared to their peers [59,60,61,62,63,64,65,66,67,68]. This is caused by their reduced rate of help-seeking and lower utilisation of regular healthcare offers [62,69]. Consequently, paediatric services have to give stronger consideration to influence of cultural values as well as to migration-related factors, in order to avoid misdiagnosis and the resulting higher costs caused by late treatment interventions [70,71]. A study by Donath et al. [72] showed that migration background could be a risk factor for higher rates of suicide attempts, especially in those originating from Islam-dominated countries. As in our study, patients originating from Turkey and Bosnia are predominantly (Sunni) Muslim; thus, religious background might be more crucial than other cultural differences.

### 4.2. Gender Differences among Natives and Migrants

In the present study sample, Turkish patients most frequently received acute treatment after a suicide attempt, but this was not the case for Austrian and BCS patients. While females with Turkish and BCS background were treated most frequently for suicide attempts, female Austrians were most likely to receive treatment at the emergency outpatient clinic for acute stress disorders. Supporting our results, preceding studies showed that the rate of suicidal behaviour among migrant females was higher, compared to their indigenous same-sex peers [44,73] and their fellow countrymen [74]. Female migrants commonly experience more migration-induced burdens [75] and their use of psychological treatment is less than their fellow countrymen [76]. Similarly, Turkish males were most frequently treated because of an attempted suicide, whereas Austrian, as well as BCS males, were most frequently treated due to behavioural disorders. In general, Turkish patients had a more than 2 times higher risk for presentation, due to suicide attempts, as compared to other patient groups. The study of van Bergen et al. [77] showed that a stressful family environment caused by different values of the society of origin and host societies might lead to increased suicidal behaviour among young migrants. In our study sample, the cultural and religious differences might be highest among Turkish families, as they mainly originate from rural areas of Turkey, and mostly maintained their traditional life-style after migration to Austria [4,11].

In line with similar studies, the number of female suicide attempts [78,79,80,81,82] as well as self-harm cases [83,84] also exceeded that of males in the present study. On the other hand, only females with a migration background had higher rates of suicidal ideation than their ethnic peers, but this was not the case among Austrian patients.

The results of the present study showed that among migrant patients, suicidal ideation was much lower compared to Austrian patients, but it was the complete opposite concerning suicide attempts.

Supporting our results, previous studies showed that migrants are overrepresented in acute psychiatry; they mostly have a referral when they are in crisis [55].

Moreover, according to the “Integrated Motivational-Volitional Model” the volitional phase factors such as impulsivity, access to means, exposure to suicide, etc. (see Introduction section) are crucial for transforming suicidal ideations to suicide attempts. Therefore, in the study by Branley-Bell et al. [10], suicide attempters varied from those having ideation in the volitional phase in different factors. Our results showed notable differences in percentages between Austrian and migrant patients in suicidal ideation and suicide attempts, which might be explained by the factors of the volitional phase of the IMV model, as described above and in the Introduction section. Furthermore, the higher rates of suicide attempts among migrants, compared to the natives was mostly due to impulsiveness in crisis situations [85].

### 4.3. Transcultural Differences in Risk Factors and Triggering Reasons

The results of the present study showed that the risk for suicide attempts among Turkish-speaking children and adolescents was more than 2 times higher than other patient groups. Noticeable transcultural differences among triggers for suicide attempts were also observed. In the whole study sample, as well over all three study groups, suicide attempts and self-harming behaviour were triggered most frequently by intrafamilial problems, but more frequently in patients with migration background than in native peers. We assume that the incidence of intrafamilial conflicts in migrant families might be quite high, due to dissimilar attitudes between the different migrant generations [11,64,86]. Especially in traditional and authoritarian-oriented Turkish families, more intrafamilial conflicts might be observed, which might cause increased rates of suicide attempts [75]. The study of van Bergen et al. [77] showed that among Turkish-speaking adolescents living in the Netherlands, suicidal ideation due to intrafamilial problems was higher compared to native peers and other minority peers. Migrant parents most commonly accentuate the culture of origin, in contrary, their children mostly are influenced by the host culture, which might increase intrafamilial conflicts [11].

As already mentioned, the results of the present study showed significant gender differences in triggering factors for attempted suicide among patients with a migration background, but this was not so among Austrians. However, no gender differences in triggering factors for self-harming behaviour between the study groups were observed.

Certain factors increased the risk of suicide attempts and of self-harming, which were different among the study groups. While relationship/separation crises triggered suicide attempts more often among BCS children/adolescents and Turkish patients, as compared to natives, the suicide attempts of Austrian patients were triggered mostly by school problems and psychological disorders. In Turkish children and adolescents, the risk for a suicide attempt was increased about 6 times when they reported problems with their peers. Furthermore, compared to other patient groups, the risk for self-harming behaviour among Turkish-speaking children and adolescents was 3 times higher, if they had problems at school. This was probably because migrant youths have to manage diverse additional school-related challenges [87,88,89].

## 5. Conclusions

The results of the present study showed that migrants usually only seek help very late, if a situation has already reached a critical level, and not early, when an intervention would be more efficient. We assume that various structural, political, and socio-cultural factors might in general create barriers for migrants in the use of the healthcare system [90], contributing to these delays. Therefore, language [91] and culture sensitive healthcare measures for migrants are required, but in Austria, they have so far been insufficiently implemented to increase the timely utilisation of the healthcare offered [92]. Equal access to healthcare is universally accepted as part of the human rights system [93].

Furthermore, the results of the present study showed that transcultural differences in risk factors and triggering reasons of suicide attempts and self-harming behaviour, especially peer-related as well as school-related problems need to be considered in preventive measures, such as psycho-education programs, to establish and increase awareness and sensitivity to this topic. These culture-adapted measures could be expected to lead to a decisive reduction of suicide attempts and self-harming behaviour among youths. Additionally, school staff and families might recognize the specific warning signs and risk groups before a crisis arises, so that they could refer the pupils for professional help much earlier. [94]. Training of health professionals in transcultural competence should also include consideration of these differences [95], and in contrast to a common focus on physical illnesses [92], evidence-based mental health in migrant youth, requires more consideration and further research [71,96], especially with regards to the complex interactions between generations and the process of adapting to the host culture [97].

### Limitations

As the present study was based on a retrospective analysis, using only available data retrieved from hospital records, data on socio-demographic features and risk factors, were frequently incomplete, possibly due to a focus on emergency treatment priorities or barriers in more prominent communication in such situations.In the present study, we analysed Turkish-speaking and BCS-speaking children and adolescents with a migration background, living in Austria. Therefore, our results cannot be generalized to other groups of migrants living in Austria.Furthermore, Turkish-speaking children and adolescents might have different ethnic backgrounds such as Kurdish, but in the present study the religion or ethnic origin of the patients was not a part of the record. As families of Kurdish or Turkish backgrounds naturally have different religious and cultural norms, this fact might lead to bias in our results. Bosnians are mainly Muslim, Serbs orthodox, and again might have very different backgrounds relating to experiences of war or religious factors. These potential differences again could not be explored, due to the limitations of the available emergency room records.

## Figures and Tables

**Table 1 ijerph-17-06498-t001:** Study sample.

Gender	Austrian Patients		Bosnian/Croatian/Serbian-Speaking Patients		Turkish-Speaking Patients		Total	
	*n*	%	*n*	%	*n*	%	*n*	%
Female	451	56.4	103	63.2	93	71.5	647	59.2
Male	349	43.6	60	36.8	37	28.5	446	40.8
Total	800	100	163	100	130	100	1093	100

**Table 2 ijerph-17-06498-t002:** Referral reasons in the different patient groups at admission time.

Referral Reasons	Austrian Patients (*n* = 800)		Bosnian/Croatian/Serbian-Speaking Patients (*n* = 163)		Turkish- Speaking Patients (*n* = 130)		Total (*n* = 1093)	
	*n*	%	*n*	%	*n*	%	*n*	%
Psychosis	9	1.1	3	1.8	2	1.5	14	1.3
Alcohol intoxication	12	1.5	0	-	0	-	12	1.1
Drug intoxication	28	3.5	8	4.9	0	-	36	3.3
Suicide attempt	88	11	27	16.6	30	23.1	145	13.3
Suicidal ideation	96	12	7	4.3	7	5.4	110	10.1
Self-harming behaviour	35	4.4	6	3.7	6	4.6	47	4.3
Acute stress disorder	167	20.9	31	19	24	18.5	222	20.3
Depressive episode	55	6.9	5	3.1	16	12.3	76	7
Anxiety and Panic disorder	78	9.8	16	9.8	17	13.1	111	10.2
Child abuse	2	0.2	2	1.2	1	0.8	5	0.5
Eating disorder	44	5.5	8	4.9	2	1.5	54	4.9
Pain	37	4.6	13	8	13	10	63	5.8
Behavioural disorder	118	14.8	24	14.7	10	7.7	152	13.9
School refusal	17	2.1	8	4.9	0	-	25	2.3
Sexual abuse	14	1.8	4	2.5	2	1.5	20	1.8
Personality disorders	0	-	1	0.6	0	-	1	0.1
Total	800	100	163	100	130	100	1093	100

**Table 3 ijerph-17-06498-t003:** Triggering factors for the suicide attempts and self-harming behaviour of three patient groups.

Triggering Factors	Austrian Patients (*n* = 123)				Bosnian/Croatian/Serbian -Speaking Patients (*n* = 33)				Turkish- Speaking Patients (*n* = 36)				Total (*n* = 192)			
	SA		SH		SA		SH		SA		SH		SA		SH	
	*n*	%	*n*	%	*n*	%	*n*	%	*n*	%	*n*	%	*n*	%	*n*	%
Intrafamilial problems	34	38.6	18	51.4	15	55.6	3	50	17	56.7	3	50	66	45.5	24	51.1
Relationship/separation crises	15	17	4	11.4	9	33.3	1	16.7	6	20	0	-	30	20.7	5	10.6
Trauma	2	2.3	2	5.7	0	-	0	-	0	-	0	-	2	1.4	2	4.3
Problems at school	16	18.2	6	17.1	1	3.7	1	16.7	2	6.7	3	50	19	13.1	10	21.3
Psychological disorder	14	15.9	4	11.4	1	3.7	1	16.7	0	-	0	-	15	10.3	5	10.6
Problems with peers	2	2.3	0	-	1	3.7	0	-	2	6.7	0	-	5	3.4	0	-
Burdens due to physical disorders	0	-	1	2.9	0	-	0	-	1	3.3	0	-	1	0.7	1	2.1
Unknown reason	5	5.7	0	-	0	-	0	-	2	6.7	0	-	7	4.8	0	-
Total	88	100	35	100	27	100	6	100	30	100	6	100	145	100	47	100

SA = suicide attempt, SH = self-harming behaviour.

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
