# Peer review of "Transcultural Differences in Risk Factors and in Triggering Reasons of Suicidal and Self-Harming Behaviour in Young People with and without a Migration Background"

_ijerph, 2020, doi:10.3390/ijerph17186498_

Round 1

Reviewer 1 Report

Thank you for the edits and clarifications.  I have no further suggestions for editing.

Author Response

Response to Reviewer Comments

Comment:Thank you for the edits and clarifications. I have no further suggestions for editing.

Response: Thank you very much for the statement.

Reviewer 2 Report

  • Please specify more clearly, at the end of the Introduction section and according to the Result and Discussion sections, the aim of the work and the main hypotheses to be verified.
  • The Introduction section is too long! Please condense it!
  • The Discussion section is also too long! Please condense it!
  • The Conclusion section is also too long! Please condense it and discuss what are the main aspects, suggested by your data, useful for the prevention and therapy of suicidal and self-harming behavior.
  • A fine/moderate check is still necessary by a native english speaker reviewer.

Author Response

Response to Reviewer Comments:

Comment: Please specify more clearly, at the end of the Introduction section and according to the Result and Discussion sections, the aim of the work and the main hypotheses to be verified.

Response:Thank you for the suggestion.As recommended, we specified more clearly at the end of the Introduction section and according to the Result and Discussion sections, the aim of the work and the main hypotheses to be verified. Page 3, lines 103-115

Comment: The Introduction section is too long! Please condense it!

Response: Thank you for the suggestion.As recommended, we substantially revised the introduction section and condensed it noticeably.

Page 2, lines 74-80, Page 3, lines 89-102 

Comment: The Discussion section is also too long! Please condense it!

Response:Thank you for the suggestion.As recommended, we substantially revised the discussion section and condensed it noticeably.

Page 8, 289-292, Page 8, 300-307, Page 9, 323-328

Page 9, 333-334, Page 9, 348-365, Page 9-10, 366-374

Comment: The Conclusion section is also too long! Please condense it and discuss what are the main aspects, suggested by your data, useful for the prevention and therapy of suicidal and self-harming behavior.

Response: Thank you for the suggestion. As recommended, we substantially revised the conclusion section. We condensed it and discussed the main aspects.

Page 10, lines 376-394 

Comment: A fine/moderate check is still necessary by a native english speaker reviewer.

Response:Thank you for the suggestion. We edited the language of the manuscript and had the whole manuscript proofread by a native speaker. The edited parts in the manuscript are marked in red. 

Reviewer 3 Report

This is an interesting study on an important topic.  Suicide is a leading cause of death around the world, understanding issues and environmental contributions are of importance to global health.

I feel that responses to reviewer comments are satisfactory.

A minor point remains with regard to language, "commit" is not generally advisable to use in reference to death by suicide and can be detrimental to ethical clarity on the issue of suicide and stigma.  This is a minor concern and it is only recommended that the authors change it.

Author Response

Response to Reviewer Comments

Comment: This is an interesting study on an important topic. Suicide is a leading cause of death around the world, understanding issues and environmental contributions are of importance to global health. I feel that responses to reviewer comments are satisfactory.

Response: Thank you very much for the statement.

Comment: A minor point remains with regard to language, "commit" is not generally advisable to use in reference to death by suicide and can be detrimental to ethical clarity on the issue of suicide and stigma. This is a minor concern and it is only recommended that the authors change it.

Response: Thank you for the suggestion. As recommended, we avoided the term “commit” and have deleted it in the whole manuscript. Page 2, lines 70-71

Reviewer 4 Report

Overall summary

The information that is presented in this manuscript suggests that there could be transcultural differences in risk factors and in triggering reasons of suicidal and self-harming behaviour in children and adolescents with and without migration background.

General comments

Although the article is interesting, the following concerns should be addressed prior to this paper is considered for publication.

1. The introduction should be rewritten. Authors include too much information which is not necessary to understand the manuscript, and furthermore, sometimes information is not clearly organised, mainly the subsection 1.4. “transcultural risk factors”; thus, it is difficult to follow the story line. Authors must remember that introductions are not exhaustive reviews of the extant literature surrounding the topic of study. Likewise, the introduction of the present study has too much references (90!) I wonder if 10 references are necessary in order to say “genetic, biological, psychiactric, psychological, social and cultural factors can influence suicidal as well as self-harming behaviour” (lines 72-73); or… if the authors need 5 references in order to say “self-haring behaviour may be associated with bullying” (lines 83, 85). Please, if it is possible, avoid the self-citation (references 4, 11-13, 19-23, 77-79, 128, 130, 131, 134).

2. In relation to the section “material and methods”:

a) the authors must include the years when the study was carried out. Saying “over a three-year period”(line 144) it is not enough because the study protocol was approved by the Ethics Committee in 2010 (ten years ago).

b) In the subsection 2.2. “Procedures and measures”, although the authors say “we analysed different risk factors, as identified in the earlier research outlined before”, it would be better to include the risk factors that may influence the suicidal and self-harmming behaviour which were analysed, in the same that it was done for the socio-demographic data (lines 182-184).

c) Please, include the significance level used in this research.

d) It is not clear if the authors did a bivariate analysis in order to determine if the differences observed among Turkish-speaking patients, BCS-speaking patients and Austrian patients were or not significative. This information is not included in the section “material and methods”, however some sentences from the section “results” (e.g. lines 280-283) suggest that it was done.

3. Despite in the section “materials and methods” is said “our results are based on sometimes incomplete sets of data” (lines 189-190) and that in the section “discussion” is said “in the hospital records, the percentage of missing data among children and adolescents with migration background was higher in comparison to their indigenous peers” (lines 326-327), nothing is mentioned about it in the section “results”. Please, include this information in section “results” in order to assess the potential bias resulting from missing data. This is very important taking into account that the percentage of missing data was different between minors with migration background and their indigenous peers.

4. According to the objective of the study, the abstract must include not only results about risk factors/triggering reasons for suicide attempts, but also for self-harming behaviour. Furthermore, the results included in the abstract should summarize the results of the study, and not include only the results of the subsection 3.7. “Impact of risk factors on suicide attempt and self-harming behaviour”.

5. Odds ratios (OR) must be expressed together with their confidence intervals. They allow to estimate the precision of the OR.

6. The subsection 3.3 “Gender differences in referral reasons among the different patient groups” needs a table to make the results clearer. A only table with the distribution by gender could be used for the subsections 3.2 and 3.3.

7. In the subsection 3.5 “Transcultural differences in risk factors”, information about professional occupation of mothers was included; was the professional occupation of fathers analyzed? Likewise, according to the results included in the subsection 3.6. “Impact of Risk factors on Acute treatment in our study”, please, include in the subsection 3.5. information about physical illness or chronic disease of the patient, tobacco consumption, and acts of violence.

Other comments:

1. Line 49. Change “,,” by: “

2. Line 57. A “)” is missing after “cause suicide”.

3. Why the Table 3 was referenced in line 305?

4. The title of the subsection 3.7 must be changed because this subsection addresses the impact of triggering factors on suicide attempt and self-harming behaviour, not the impact of risk factors on suicide attempt and self-harming behaviour.

5. Lines 483-484. The link does not work.

6. Line 282. The “:” after “(p=0.000)” makes no sense.

7. The authors must include a reference for the following sentencesMore than half of...Austria had a migration background” (lines 334-337) and “Nevertheless, as… to the native population (lines 123-126).

8. The table 2 has not been mentioned in the main text.

9. For embedded citations in the text, the years are not necessary (e.g. lines 61, 93, 96, 101, 407…).

10. Line 430. The conclusion “the results of the present study show that migrants usually only seek help in crisis” is not true. Maybe you can change “show” by “suggest”.

11. The references list must be reviewed. It has some mistakes. For example: page spreads instead of abbreviated page numbers must be used.

Round 2

Reviewer 4 Report

In general, the authors have addressed the changes which were previously suggested. However, it is a pitty that: i) they have not included all the specific risk factors that may influence the suicidal and self-harming behaviour which were analysed (page 4, line 159); ii) the Table 2 does not include the distribution by gender; iii) the authors have not done a bivariate analysis in order to determine if the differences observed among Turkish-speaking patients, BCS-speaking patients and Austrian patients were or not significative.

This manuscript is a resubmission of an earlier submission. The following is a list of the peer review reports and author responses from that submission.

Round 1

Reviewer 1 Report

This study analyzes the risk factors of suicidal behaviours in young people with and without migration background for patients visiting a psychiatric emergency room in Vienna. I think the topic of this study to identify whether the risk of suicide differs depending on the migration background for young people, and whether the factors that trigger the suicide action are different is meaningful, but I think that the value of this study is limited because of the following reasons.

1) This study targets only young people who visit a psychiatric emergency room, and it is questionable how much the results of this study can reflect the actual situation. For example, more than half of the subjects were immigrants while about a quarter of Austria had immigration background. But I question the possibility of a higher percentage of immigrants living near the hospital.
2) Comparing the characteristics of suicide attempters and those of other patients visiting psychiatric emergency room is not proper to examine the risk factors of suicidal behaviours because other patients were also in psychiatric acute stage. 
3) The number of study subjects in method section and result section does not match.
4) The method of odds ratio calculation is not clear. Multivariate analysis needs to be implemented using datasets for each individual.
5) The overall writings require short and logical description.

Reviewer 2 Report

Thank you for the chance to review this very important work.

Below are my suggestions for clarifications/changes

abstract lines 33-34: I would recommend rephrasing "Therefore mental health professionals need transcultural competences for an optimal treatment of them." to "Therefore, mental health professionals need transcultural competence to provide optimal treatment of children with a migration background" or something similar

lines 111-112: "Therefore migrant population often has a higher rate of psychological problems" should be rephrased to "Therefore migrant populations often have a higher rate of psychological problems

lines 121-123: rephrase, awkward grammar 

lines 201-202: would you consider 4.6% to be significantly more than 4.4%, I would conclude that the number of patients seeking treatment for self-harm was similar between these groups

lines 254-255: please define "compulsory school educated"

line 258: please define "high educated"

line 313: change "...with migration background have often a reduced rate..." to "...with migration background often have a reduced rate...."

line 321: change "not sufficiently" to "insufficient"

lines 342, 356 and 381: remove the "s" from the end of "patients" so it reads "patient groups"

line 357: change "...suicide attempts showed also marked..." to "...suicide attempts also showed marked..."

line 358: change "mostly" to "most"

Reviewer 3 Report

I can only imagine how difficult it is to produce a scientific piece of writing in something other than your first language. This is certainly a lot better than anything I could attempt. However, the style and grammatical mistakes in the English make this a difficult to follow manuscript in places. I believe that the manuscript needs extensive editing by someone with greater English writing skills. My difficulty in following the English has meant I am unable to provide as thorough a review as I would have aimed to.

I do also have a few other suggestions where I believe improvements could be made:

Lines 50-55 - discussion of the Integrated Motivational-Volitional Model: First, the model is inadequately explained. Second, it is not clear to me how the model relates to the rest of the paper. There is no reference to it in the discussion of results or conclusion. I have the impression that there was a perceived need to reference a theory and this was thrown in.

Table 3: SA and SH are not defined

Missing from the discussion is the the difference in difference between the Austrian and the migrant patients in regard suicide ideation. The percentage of migrants with suicide ideation is much lower than that of the Austrians. Hence it would appear that migrants are more likely present when they are in crisis. Possibly this is where a discussion of the Integrated Motivational-Volitional Model is relevant.

Lines 354-385: The lower citing of intrafamilial problems in the Austrian patients may be because their parents can understand what they are saying. Hence they might be more likely to say they are having problems with school. There could be other cultural factors behind what it is acceptable for them to report as the cause for their attempt. I think a more nuanced consideration of what might be reported in relation to culture and ability of parents to understand what is being reported is needed.

Conclusion: I find the conclusion rather vague and fails to say anything of any substance. I'm left wondering what this paper is trying to contribute.